# Intracellular Life Cycle Kinetics of SARS-CoV-2 Predicted Using Mathematical Modelling

**DOI:** 10.3390/v13091735

**Published:** 2021-08-31

**Authors:** Dmitry Grebennikov, Ekaterina Kholodareva, Igor Sazonov, Antonina Karsonova, Andreas Meyerhans, Gennady Bocharov

**Affiliations:** 1Marchuk Institute of Numerical Mathematics, Russian Academy of Sciences (INM RAS), 119333 Moscow, Russia; kholodareva.ea@phystech.edu; 2Moscow Center for Fundamental and Applied Mathematics at INM RAS, 119333 Moscow, Russia; 3World-Class Research Center “Digital Biodesign and Personalized Healthcare”, Sechenov First Moscow State Medical University, 119991 Moscow, Russia; 4Moscow Institute of Physics and Technology (National Research University), Dolgoprudny, 141701 Moscow Oblast, Russia; 5College of Engineering, Swansea University, Bay Campus, Fabian Way, Swansea SA1 8EN, UK; i.sazonov@swansea.ac.uk; 6Department of Clinical Immunology and Allergology, Sechenov First Moscow State Medical University, 119991 Moscow, Russia; karsonova@gmail.com; 7Infection Biology Laboratory, Universitat Pompeu Fabra, 08003 Barcelona, Spain; andreas.meyerhans@upf.edu; 8ICREA, Pg. Lluis Companys 23, 08010 Barcelona, Spain; 9Institute of Computer Science and Mathematical Modelling, Sechenov First Moscow State Medical University, 119991 Moscow, Russia

**Keywords:** SARS-CoV-2, intracellular replication, mathematical model, sensitivity analysis, targets for drugs

## Abstract

SARS-CoV-2 infection represents a global threat to human health. Various approaches were employed to reveal the pathogenetic mechanisms of COVID-19. Mathematical and computational modelling is a powerful tool to describe and analyze the infection dynamics in relation to a plethora of processes contributing to the observed disease phenotypes. In our study here, we formulate and calibrate a deterministic model of the SARS-CoV-2 life cycle. It provides a kinetic description of the major replication stages of SARS-CoV-2. Sensitivity analysis of the net viral progeny with respect to model parameters enables the identification of the life cycle stages that have the strongest impact on viral replication. These three most influential parameters are (i) degradation rate of positive sense vRNAs in cytoplasm (negative effect), (ii) threshold number of non-structural proteins enhancing vRNA transcription (negative effect), and (iii) translation rate of non-structural proteins (positive effect). The results of our analysis could be used for guiding the search for antiviral drug targets to combat SARS-CoV-2 infection.

## 1. Introduction

Human infection with SARS-CoV-2 presents a tremendous health problem. The within-host infection characteristics are characterized by extreme variability of the disease course ranging from asymptomatic infections to severe forms of COVID-19 with lethal outcomes [1]. This spectrum of pathogenicity is a result of the interaction of numerous processes and factors on multiple levels of realization: the single cell, tissues and organs, and the organism’s physiology [2]. The SARS-CoV-2 infection dynamics in a human organism is determined by the kinetics of infection of cells which express viral receptors (e.g., ACE2), by the activation of the intracellular defense, and by systemic immunophysiological reactions. The corresponding “virus–human organism” is a multiscale, multicomponent dynamical system. A comprehensive study of such a system requires the development of mathematical models that integrate the underlying biophysical, biochemical, and physiological processes [3]. Although a few mathematical models have been recently proposed to describe the within-host kinetics of SARS-CoV-2 infection [4,5,6,7,8,9], the degree of a mechanistic process resolution remains to be greatly enhanced.

From a scientific point of view, the intracellular replication of the viruses is realized as an intertwined set of biochemical reactions and biophysical transport processes. Being summarized in a schematic form, the theoretical abstraction of the system provides a conceptual platform for deriving a mathematical model. The mathematical descriptions can further be specified on the basis of the law of mass action, enzyme kinetics, diffusion laws, etc, following deterministic, stochastic, or hybrid approaches. The derived mathematical models which bear a direct resemblance to and coordination with the underling chemical and physical processes are considered as mechanistic models versus empirical (e.g., statistical) or black-box (e.g., neural-networks) type mathematical models. The mechanistic models provide a descriptive and analytical tool for studying the virus life cycle regulation and predicting its response to structural or parametric perturbations. In virology, the mechanistic models have been successfully developed and applied to study, in a quantitative way, the intracellular replication of HIV-1 [10], hepatitis B virus [11], influenza A virus [12], hepatitis C virus [13], and poliovirus [14].

The dynamic view on the balance between virus expansion and development of antiviral immune responses suggests that the kinetics of virus growth is one of the major determinants of the infection trajectory [15]. Existing mathematical models of SARS-CoV-2 infection consider the virus growth at the cell population level, ignoring details of the virus life cycle. The infection of permissive cells by the virus results in a number of dramatic changes of the infected cell physiology, such as modification of host protein synthesis [16], inhibition of the innate immune responses [17], and induction of programmed cell death [18]. In this study, we formulate a mathematical model of SARS-CoV-2 replication in infected cells. The development of the model follows a deterministic approach, similar to [10], in order to focus on the calibration of model parameters to reproduce some reference kinetics of the virus life cycle steps. By applying a sensitivity analysis of the model, we rank the model parameters according to their impact on the net secretion of virions by the infected cell. The most influential parameters can be considered as prospective targets for available antiviral drugs or novel treatments.

## 2. Methods

### 2.1. Model Development and Calibration

The kinetics of the corresponding biochemical reactions is described in the deterministic mathematical model introduced in Section 3. The system of ordinary differential equations (ODEs) is formulated using the law of mass action and Michaelis–Menten parameterization. Note that the deterministic version provides an initial step toward a stochastic description of virus replication.

We use a Michaelis–Menten-type description for the nucleocapsid formation and viral assembly processes to represent the following aspects of virus replication: (i) purified coronavirus virions containing mainly full-genome length viral RNAs, i.e., there is no saturation in the kinetics with respect to viral RNA [18], (ii) structural proteins (N and M) required for virion morphogenesis and recruitment of the virion structural components to the assembly site [1] scale this rate so that enough proteins must be present above a certain level to reach half the maximal rate of the ribonucleoprotein formation and virion assembly, and (iii) the least abundant structural proteins represent the kinetic bottleneck in the overall progeny production.

A major step in the mathematical model formulation is the estimation of parameters appearing on the right-hand sides of the model equations. The maximum likelihood approach represents a general framework to address the parameter estimation problem. However, the sparsity of the available kinetic data on the intracellular life cycle of SARS-CoV-2 motivates the implementation of the parameter estimation procedure, which we refer to as “model calibration”. It stands for an iterative process of parameter guessing and constraining the choice progressively by requiring consistency with all available data. In this study, these data mainly consist of previously estimated numerical values and/or numerical ranges of the molecular species variables at key time points, such as the beginning and end of the infection cycle, or of intermediate steps, as summarized in Table 1.

**Table 1 viruses-13-01735-t001:** Time-dependent variables of the mathematical model characterizing the SARS-CoV-2 life cycle.

Variable	Meaning	Quantitative Characteristics
$\left[V_{free}\right]$	number of free virions outside the cell membrane	10
$\left[V_{bound}\right]$	number of virions bound to ACE2 and activated by TMPRSS2	1–10
$\left[V_{endosome}\right]$	number of virions in endosomes	1–10
$\left[gRNA_{\left(+\right)}\right]$	single strand positive sense genomic RNA	1–5
[NSP]	population of non-structural proteins	−
$\left[gRNA_{\left(-\right)}\right]$	negative sense genomic and subgenomic RNAs	10
[gRNA]	positive sense genomic and subgenomic RNAs	10,000
[SP]	total number of structural proteins S+M+E per virion	2000∈(1125,2230) [19,20,21]
[N]	*N* proteins per virion [N]	456 [21]; 1465∈(730,2200) [19]
[N-gRNA]	ribonucleocapsid molecules	−
$\left[V_{assembled}\right]$	assembled virions in endosomes	−
$\left[V_{released}\right]$	virus burst size	10–10,000 virions in 7 to 24 h [2,22,23]

### 2.2. Model Validation

To validate the calibrated model, we compared some predicted quantities with available experimental data, namely, (i) the kinetics of positive- to negative-sense vRNA ratio in MHV-infected cells [24], and (ii) the kinetics of SARS-CoV-2 replication in the cell cultures of Vero E6 cells [23,25]. As we cannot directly infer the absolute values of the number of released virions by an infected cell just because there are no such single-cell experiments, we used the relative measure of fold changes of titer levels in cell culture from the start of their exponential growth. This data was compared with the fold changes of $\left[V_{released}\right]\left(t\right)$ starting from the moment $t_{s}$ at which $\left[V_{released}\right]\left(t_{s}\right)=1$.

The experimental data presented in [24] characterize the kinetics of virus transcription and translation of cells infected with Murine coronavirus (MHV-A59). In particular, the ribosome profiling technique was used to generate high precision data on the production of the positive and negative-sense genomic and subgenomic viral RNAs. The overall kinetics of SARS-CoV-2 in Vero E6 cells is studied in [23]. To this end, one-step growth (MOI = 5) of three recombinant (i.e., icSARS-CoV-Urbani, icSARS-CoV-GFP, icSARS-CoV-nLuc) and one clinical strain WA1 was followed. The data on the kinetics of virus titers in the supernatants of cell cultures (PFU/mL) show that the growth phase until reaching the plateau takes about 12 to 19 h and is similar to all the above CoV variants. A similar study of SARS-CoV-2 replication features such as growth kinetics, virus titers, analysis of transcription, and translation in Vero E6 cells is presented in [25]. The virus isolate SARS-CoV-2 Australia/VIC01/2020 was used for infection of cells at MOI = 3. Intracellular viral RNA, protein synthesis and release of infectious viral progeny (by plaque assay) were quantified. The initial guess for the mathematical model parameters was specified using data from a broad spectrum of publications covering the structural and genetic properties of the SARS-CoV-2, the coronavirus transcription and translation, and the turnover of proteins and RNA in cells with the relevant references presented in Section 3.

### 2.3. Parameter Uncertainty Analysis

Through the process of model calibration, the point estimates of parameter values were determined, which were used in main simulations and are discussed in Section 3. For most of the model parameters, we were able to estimate the ranges of biologically plausible values based on literature data. For other parameters, we derived their ranges applying the parameter uncertainty analysis. To this end, we iteratively adjusted their parameter ranges so that they would include their point estimates while the uncertainty of the model output (progeny release kinetics) would be confined within the range from 10 to 10,000 virions, as specified in Table 1. For some parameters with wide literature-based ranges, we further narrowed them down to restrict the output uncertainty. These three categories of parameter ranges (based on literature analysis, based on uncertainty analysis, based on both types of analysis), as well as parameters which are fixed ad hoc, are reviewed in Section 3 accordingly. To quantify the output uncertainty, we employed the Latin hypercube sampling method to randomly sample n=10,000 combinations of parameters from their respective ranges. Then, the median and 5–95% confidence regions of the obtained ensemble of $\left[V_{released}\right]\left(t\right)$ trajectories were computed.

### 2.4. Sensitivity Analysis

Let u(t,p) be the model solution of the ODE initial value problem $\frac{du}{dt}=f\left(t,u,p\right)$, $u\left(0\right)=u_{0}$. In this study, we apply the methods of local sensitivity analysis to identify the parameters (and respective biochemical processes) that have the most strong impact on the characteristics of interest Φ(u(p)), e.g., the cumulative number of released virions
$$\Phi_{progeny}=\int_{0}^{T}k_{release}\left[V_{assembled}\right]dt,$$
the area under the curve for $\left[V_{released}\right]\left(t\right)$
$$\Phi_{progeny}^{AUC}\left(p\right)=\int_{0}^{T}\left[V_{released}\right]dt,$$
or the distance between the model output g(u(p)) and the experimental data $g_{exp}\left(t_{i}\right)$
$$\Phi =\sum_{i}\;||g\left(u\left(t_{i}\right)\right)-g_{exp}\left(t_{i}\right){\left|\right|}^{2}.$$

The sensitivity index of model parameter *p* is defined as $s_{p}=\frac{d\Phi (p)}{dp}$. It indicates the influence of parameter variation on the output value Φ. To compare sensitivities and rank parameters by their impact, the sensitivity indices are usually normalized by parameter values: ${\hat{s}}_{p}=p\frac{d\Phi (p)}{dp}$. The sensitivity indices can be computed by solving the forward sensitivity ODE system or by solving the adjoint problem [26]. We follow the adjoint-based approach as previously described [10,26].

### 2.5. Software

The following packages in Julia language (https://julialang.org, accessed on 12 May 2021) were used to simulate and analyze the model: DifferentialEquations.jl (accessed on 12 May 2021) for numerical solution of the model, QuasiMonteCarlo.jl (accessed on 12 May 2021) for parameter uncertainty analysis, DiffEqSensitivity.jl (accessed on 12 May 2021) for local sensitivity analysis, PyPlot.jl (accessed on 12 May 2021) for visualizations. The scripts used to simulate and analyze the model are provided in the Appendix A.

## 3. Results

We follow the latest view of the SARS-CoV-2 life cycle summarized in [1,17,18,27]. The key steps include: (i) cell entry, (ii) genome transcription and replication, (iii) translation of structural and accessory proteins, and (iv) assembly and release of virions (Figure 1). The set of time-dependent molecular species described in the model is listed in Table 1 with some quantitative characteristics of their abundance.

### 3.1. Mathematical Model of Intracellular SARS-CoV-2 Replication

#### 3.1.1. Cell Entry

The entry stage of SARS-CoV-2 is split into three steps as represented in Figure 1:binding of the receptor-binding domain (RBD) of the viral S protein to the ACE2 receptor,priming by host cell surface protease TMPRSS2,fusion at the cellular or endosomal membrane followed by release and uncoating of the viral genomic RNA.

Binding of the virion to the cellular transmembrane protein ACE2, and entry and release of the viral RNA into the host cell are described by equations specifying the rates of changes of free-, receptor-bound, and fused virions, as well as the viral RNA genome in the cytoplasm:(1)$$\begin{array}{cc} \frac{d[V_{free}]}{dt} & =-k_{bind}\left[V_{free}\right]-d_{V}\left[V_{free}\right]+k_{diss}\left[V_{bound}\right] \end{array}$$(2)$$\begin{array}{cc} \frac{d[V_{bound}]}{dt} & =k_{bind}\left[V_{free}\right]-\left(k_{fuse}+k_{diss}+d_{V}\right)\left[V_{bound}\right] \end{array}$$(3)$$\begin{array}{cc} \frac{d[V_{endosome}]}{dt} & =k_{fuse}\left[V_{bound}\right]-\left(k_{uncoat}+d_{endosome}\right)\left[V_{endosome}\right] \end{array}$$(4)$$\begin{array}{cc} \frac{d[gRNA_{\left(+\right)}]}{dt} & =k_{uncoat}\left[V_{endosome}\right]-d_{gRNA}\left[gRNA_{\left(+\right)}\right]. \end{array}$$

Here, $\left[V_{free}\right]$ is the number of free virions outside the cell membrane, $\left[V_{bound}\right]$ is the number of virions bound to ACE2 and activated by TMPRSS2, $\left[V_{endosome}\right]$ is the number of virions in endosomes, and $\left[gRNA_{\left(+\right)}\right]$ is the number of ss-positive sense genomic RNA. The respective parameters of the above equations are described in Table 2. In estimating the degradation rate of virions in endosomes, we followed the assumptions presented for modelling IAV infection [12], i.e., about 50% fail to release the viral genome. This gives $d_{endosome}=0.06$ h${}^{-1}$.

#### 3.1.2. Genome Transcription and Replication

The SARS-CoV-2 virion consists of about a 30 kb strand of positive sense RNA coated with *N* protein and covered by a lipid bilayer containing spike *S*, membrane *M*, and envelope *E* proteins [18]. The model was calibrated to reproduce (i) the scale of viral proteins production corresponding to about 10 to 10,000 infectious virions per cell, (ii) the observed ratio of positive and negative sense viral RNA (genomic and subgenomic) [24], and (iii) the known ranges of the parameters of mRNA and protein turnover [28].

The released genomic RNA undergoes translation into viral polyproteins (pp1a, pp1ab) which generate via proteolysis 16 non-structural proteins (nsp1–16). They are operating to form the viral replication and transcription complex. In particular, a key step is the formation of nsp12, which encodes the RNA-dependent RNA polymerase (RdRp). The primary function of the RdRp replication complex is to generate a negative sense full-length genome and subgenomic RNAs. It has been established for MHV virus that synthesis of negative-sense RNA starts about 60 to 90 min post-infection and reaches a maximum at about 5 to 6 h [24,29]. The resulting set of negative-sense RNAs and the full-length antisense genome are working as templates for the synthesis of positive-sense genomic and subgenomic RNAs as shown in Figure 1. The total number of produced positive-sense viral genomes and subgenomic RNAs exceeds the number of negative-sense RNAs by 100 to 1000 fold [24,29].

We describe the abundance of the populations of non-structural proteins [NSP], the set of negative sense genomic and subgenomic $\left[gRNA_{\left(-\right)}\right]$, and the set of positive sense genomic and subgenomic [gRNA] with the following differential equations: (5)$$\begin{array}{cc} \frac{d[NSP]}{dt} & =k_{transl}f_{ORF1}\left[gRNA_{\left(+\right)}\right]-d_{NSP}\left[NSP\right] \end{array}$$(6)$$\begin{array}{cc} \frac{d[gRNA_{\left(-\right)}]}{dt} & =k_{tr_{\left(-\right)}}\left[gRNA_{\left(+\right)}\right]\theta_{RdRp}-d_{gRNA_{\left(-\right)}}\left[gRNA_{\left(-\right)}\right] \end{array}$$(7)$$\begin{array}{cc} \frac{d[gRNA]}{dt} & =k_{tr_{\left(+\right)}}\left[gRNA_{\left(-\right)}\right]\theta_{RdRp}-\left(k_{complex}\theta_{complex}+d_{gRNA}\right)\left[gRNA\right] \end{array}$$
where
(8)$$\theta_{RdRp}=\frac{\left[NSP\right]}{\left[NSP\right]+K_{NSP}},\;\theta_{complex}=\frac{\left[N\right]}{\left[N\right]+K_{N}}$$

It is taken into account in Equation (Equation 5) that the non-structural proteins are translated only from the released genomic RNA. Similarly, the transcription of the negative-sense genomic and subgenomic RNA described by Equation (6) is determined by the original positive-sense genomic RNA.

The translation rate $k_{transl}=45$,360 nt/mRNA h${}^{-1}$ has been estimated for coronaviruses in [24]. The length of the RNA genome coding for [NSP] proteins is about 21,000 nucleotides [30], hence $f_{ORF1}=1/21$,000. These estimates are consistent with the general range of protein synthesis rates: $\left(1,10^{4}\right)$ molecules/mRNA h${}^{-1}$ [28].

The degradation rate of NSPs $d_{NSP}=0.069$ h${}^{-1}$ is estimated as the geometric mean of the half-lives of proteins in cells. This result of our calibration procedure is consistent with the finding that the lifetime of most proteins is just a few hours [28].

The transcription rates of negative sense genomic and subgenomic RNAs is estimated to be $k_{tr_{\left(-\right)}}=3$ copies/mRNA h${}^{-1}$. This is consistent with the transcription rate of mRNAs which ranges from 1 to 100 copies per hour according to [28].

We assume the threshold for half-maximal rate of RdRp activity to be $K_{NSP}=100$ copies, taking into account that a small number of non-structural proteins is sufficient for enhancing the transcription of vRNAs.

It is known that an average half-life of mRNAs in cells of vertebrates is about 3 h [31] and ranges from 1 to 10 h [28]. Hence, the decay rate of mRNAs can be in between [0.069,0.69] h${}^{-1}$. We use here the following value $d_{gRNA}=0.2$ h${}^{-1}$. In addition, we assume a smaller value for the degradation rate of negative sense vRNAs in double-membrane vesicles, $d_{gRNA_{\left(-\right)}}=d_{gRNA}/2=0.1$ h${}^{-1}$.

Quantitative analysis of RNA polymerase elongation provides the estimate of the transcription rate to be 46,080±17,640 nt/h [32]. Thus, the basal rate of transcription is around 46,080/30,000 nt/RNA h${}^{-1}$. In infected cells, however, the overall rate of viral RNA transcription is amplified (around a 1000-fold increase) while transcription of host RNAs is largely silenced [30]. Therefore, we estimate $k_{tr_{\left(+\right)}}=1000$ copies/mRNA/h, which matches the observed ratio of positive- to negative-sense viral RNAs [24,29] (see the model validation results at the end of this Section).

The kinetics of the nucleocapsid formation (i.e., viral RNA genome coated with N protein) resulting from the binding of N proteins and gRNA is characterized by the rate constant $k_{complex}$. This can be estimated from the binding data presented in [33]. The data indicate a fast kinetics with a characteristic time of ≈20 s. Hence $k_{complex}\approx 0.4$ h${}^{-1}$, taking into account that 1 virion consists of 38 ribonucleoprotein complexes each having about 12 N proteins [21], $n_{N}=38\times 12=456$.

The kinetics of formation of the nucleocapsid condensates from N proteins and gRNA has been studied in [34,35]. The minimum concentration of N proteins necessary to form condensates has been estimated to be about 3.3μM, and the concentration at which the formation slows down to be about 11μM. These concentrations correspond to approximately 1.5 and 5 million molecules, if we take 768 fL as the mean cell volume of type II pneumocyte [2]. The estimate $K_{N}=5\times 10^{6}$ molecules implies that the saturation takes place at the number of N proteins necessary to assemble around 11,000 virions ($K_{N}\approx n_{N}\cdot 11,000$).

#### 3.1.3. Translation of Structural and Accessory Proteins

The structural proteins *S*, envelope *E*, and membrane *M* are translated from the positive sense subgenomic RNAs at the endoplasmic reticulum (ER). They are described in the model by their total abundance [SP]. They are considered to interact together and assist in forming virus-like particles and budding of new virions from the ER and Golgi compartments (ERIGC) [17]. The structural nucleocapsid protein N is translated from subgenomic RNAs by cytosolic ribosomes and can enhance the formation of virus-like particles [27]. Its key function is to create nucleocapsid [N-gRNA] by coating genomic RNAs. The number of N proteins per virion [N] in coronaviruses is estimated to range from 730 to 2200 [19]. For SARS-CoV-2, however, the estimated number of N proteins per virion is $n_{N}=38\times 12=456$ [21].

The translation rates of [N] and [SP] proteins are described by the following two equations: (9)$$\begin{array}{cc} \frac{d[N]}{dt} & =k_{transl}f_{N}\left[gRNA\right]-k_{complex}n_{N}\theta_{complex}\left[gRNA\right]-d_{N}\left[N\right] \end{array}$$(10)$$\begin{array}{cc} \frac{d[SP]}{dt} & =k_{transl}f_{SP}\left[gRNA\right]-k_{assemb}n_{SP}\theta_{assemb}\left[\mathrm{N-gRNA}\right]-d_{SP}\left[SP\right] \end{array}$$
where
(11)$$\theta_{assemb}=\frac{\left[SP\right]}{\left[SP\right]+K_{V_{\mathrm{rel}}}n_{SP}}$$

The scaling parameter $f_{N}=1/1200$ accounts for the length of the N-coding RNA which is about 1200 nucleotides (400 aa) [36]. Likewise, $f_{SP}=1/10$,000, as the estimated RNA length for the structural proteins S, E, and M is about 10,000 nucleotides.

The degradation rate of N protein $d_{N}=0.023$ h${}^{-1}$ is estimated using the database [37]. The degradation rate of the mixture of other structural proteins is evaluated to be around $d_{SP}=0.044$ h${}^{-1}$. The plausible range [0.023,0.36] h${}^{-1}$ is guessed using the half-lives of N, M, E, and S proteins being about 30 and 1.9 h, respectively, in reticulocytes, and their relative molar ratio in a virion E:S:M=1:20:300.

The composition of a single virion requires the binding of about 456 N protein molecules to each positive-sense genomic RNA, $n_{N}=456$. The total number of structural proteins S,M,E can be estimated to be in between 1125 to 2230 [19,20,21] with the reference number we used about 2000, i.e., $n_{SP}=2000$.

The scale of SARS-CoV-2 replication depends on the target cell type (e.g., bronchial, lung cells, enterocytes). The available estimates suggest that the duration of the single replication cycle ranges from 7 to 24 h with the burst size in between 10 and 10,000 virions [2,22]. Hence, we set $K_{V_{\mathrm{rel}}}=1000\in (10,10$,000). Similar to modelling studies on replication of HIV-1 [10] and IAV [12], we estimate the following range for the rate of virion assembling $k_{assemb}$ = 0.01–10 h${}^{-1}$.

#### 3.1.4. Assembly and Release of Virions

The assembly of virions requires that nucleocapsid and viral envelope glycoproteins coalesce into the same domain of the intracellular space [18]. The nucleocapsid core of the virion traffics to ERGIC and buds into ERGIC membranes covered with the structural proteins. Thus, a lipid envelope of the virion is created. The genome packaging is mediated by a packaging signal unique to genome length RNA.

As discussed above, the N proteins are key for incorporating viral RNA into viral progeny particles [38,39]. There, the N-terminal RNA-binding domain binds the RNA and the C-terminal domain via interaction with the M protein functions to anchor the ribonucleoprotein to the viral membrane [40].

The virions are assembled at the ER-Golgi compartment via encapsulating N-RNA complexes. Assembled new virions can exit the infected cell by exocytosis via the lysosomal trafficking pathway, budding, or cell death [17].

The rates of changes of the ribonucleocapsid and the assembled and released virions are described by the following equations: (12)$$\begin{array}{cc} \frac{d[\mathrm{N-gRNA}]}{dt} & =k_{complex}\theta_{complex}\left[gRNA\right]-\left(k_{assemb}\theta_{assemb}+d_{N-gRNA}\right)\left[\mathrm{N-gRNA}\right] \end{array}$$(13)$$\begin{array}{cc} \frac{d[V_{assembled}]}{dt} & =k_{assemb}\theta_{assemb}\left[\mathrm{N-gRNA}\right]-\left(k_{release}+d_{assembled}\right)\left[V_{assembled}\right] \end{array}$$(14)$$\begin{array}{cc} \frac{d[V_{released}]}{dt} & =k_{release}\left[V_{assembled}\right]-d_{V}\left[V_{released}\right] \end{array}$$

We use the estimate of $d_{gRNA}$ for $d_{N-gRNA}=0.2$ h${}^{-1}$, taking into account that the major component of ribonucleoprotein is ssRNA.

The direct measurements of the budding rate are not available yet. However, it has been shown for in vitro systems to be a very fast process with a characteristic time of about 2 s [41]. Hence, the following range is biologically plausible: $k_{release}\in \left[8,7200\right]$ h${}^{-1}$ [10].

As for the assembled virion death rate, we use the value $d_{assembled}=0.06$ h${}^{-1}$ estimated from [42], which is equal to $d_{endosome}$.

The overall list of model parameters with their reference values and permissible ranges is presented in Table 2. The solution of the model described by Equations (Equation 1)–(Equation 14) for $\left[V_{free}\right]\left(0\right)=10$ and the parameter values displayed in Table 2, is shown in Figure 2. It is consistent with data presented in [24,43].

In this study, we analyze the model behaviour at the initial condition $\left[V_{free}\right]\left(0\right)=10$. This corresponds to high MOI scenario of in vitro experiments in tissue cultures that are performed to estimate the burst size, i.e., the average number of virions produced by a single infected cell during the complete replication cycle. High MOI is typically used to ensure that every single cell gets infected and therefore only a single replication cycle occurs, resulting in a “one-step” growth dynamics of released progeny [2].

**Table 2 viruses-13-01735-t002:** Estimates of the calibrated model parameters. The parameter range categories are labelled as follows: ranges based on the analysis of (†) literature, (‡) uncertainty quantification, (†‡) both.

Parameter	Description, Units	Value	Range, Relev. Refs.
$k_{bind}$	rate of virion binding to ACE2 receptor, h${}^{-1}$	12	$\left(3.6,12\right){\;}^{†}$ [44,45]
$d_{V}$	clearance rate of extracellular virions, h${}^{-1}$	0.12	$\left(0.06,3.5\right){\;}^{†}$ [42,46,47], tuned to $\left(0.06,0.2\right){\;}^{†‡}$
$k_{diss}$	dissociation rate constant of bound virions, h${}^{-1}$	0.61	$\left(0.32,1.08\right){\;}^{†}$ [44,45]
$k_{fuse}$	fusion rate constant, h${}^{-1}$	0.5	$\left(0.33,1\right){\;}^{†}$ [48]
$k_{uncoat}$	uncoating rate constant, h${}^{-1}$	0.5	$\left(0.33,1\right){\;}^{†}$ [48]
$d_{endosome}$	degradation rate of virions in endosomes, h${}^{-1}$	0.06	[12,42], $\left(0.0001,0.12\right){\;}^{‡}$
$k_{transl}$	translation rate, nt/mRNA h${}^{-1}$	45,360	[24,28], (40,000,50,$000){\;}^{‡}$
$1/f_{ORF1}$	length of ORF1 of the RNA genome coding [NSP], nt	21,000	fixed [30]
$d_{NSP}$	degradation rate of proteins in the cell, h${}^{-1}$	0.069	$\left(0.023,0.69\right){\;}^{†}$ [28,37], tuned to $\left(0.023,0.1\right){\;}^{†‡}$
$k_{tr_{\left(-\right)}}$	transcription rate of negative sense genomic and subgenomic RNAs, copies/mRNA h${}^{-1}$	3	$\left(1,100\right){\;}^{†}$ [28], tuned to $\left(1,20\right){\;}^{†‡}$
$K_{NSP}$	threshold number of [NSP] enhancing vRNA transcription, molecules	100	$\left(10,150\right){\;}^{‡}$
$d_{gRNA}$	degradation rate of positive sense RNAs in cell, h${}^{-1}$	0.2	$\left(0.069,0.69\right){\;}^{†}$ [28,31], tuned to $\left(0.069,0.4\right){\;}^{†‡}$
$d_{gRNA_{\left(-\right)}}$	degradation rate of negative sense RNAs in double-membrane vesicles, h${}^{-1}$	0.1	$\left(0.05,0.2\right){\;}^{‡}$
$k_{tr_{\left(+\right)}}$	replication rate of positive sense RNAs, copies/mRNA/h	1000	$\left(620,1380\right){\;}^{†}$ [32]
$k_{complex}$	rate of the nucleocapsid formation [N-gRNA], h${}^{-1}$	0.4	$\left(0.02,0.4\right){\;}^{†}$ [21,33,49,50,51]
$K_{N}$	threshold number of N proteins at which nucleocapsid formation slows down, molecules	$5\times 10^{6}$	$\left(3.5,6.5\right)\times 10^{6\;†}$ [2,34,35]
$1/f_{N}$	length of RNA genome coding *N* protein, nt	1200	fixed [36]
$1/f_{SP}$	length of genome coding structural proteins S,E,M, nt	10,000	fixed [36]
$d_{N}$	degradation rate of *N* protein, h${}^{-1}$	0.023	$\left(0.023,0.069\right){\;}^{†}$ [37]
$d_{SP}$	mean degradation rate of the pool of E,S,M proteins, h${}^{-1}$	0.044	$\left(0.023,0.36\right){\;}^{†}$ [37]
$n_{SP}$	total number of structural proteins S,M,E per virion, molecules	2000	$\left(1125,2230\right){\;}^{†}$ [19,20,21]
$n_{N}$	number of *N* protein per virion, molecules	456	fixed [21]
$K_{V_{\mathrm{rel}}}$	threshold number of virions at which the virion assembly process slows down, virions	1000	(10,10,$000){\;}^{†}$ [2,22]
$k_{assemb}$	rate of virion assembling, h${}^{-1}$	1	$\left(0.01,10\right){\;}^{†}$ [10,12]
$d_{\mathrm{N-gRNA}}$	degradation rate of ribonucleoprotein, h${}^{-1}$	0.2	$\left(0.069,0.69\right){\;}^{†}$ [28,31]
$k_{release}$	rate of virion release via exocytosis, h${}^{-1}$	8	$\left(8,7200\right){\;}^{†}$ [10,41]
$d_{assembled}$	assembled virion degradation rate, h${}^{-1}$	0.06	[42], $\left(0.0001,0.12\right){\;}^{‡}$

The simulations cover the range of 24 h, which is informative for a single replication cycle setting experimental data on SARS-CoV-2 replication [2,22,23]. However, the interval can be stretched via amending the parameter values to adjust more specific data for particular cell types and infection conditions (e.g., Figure 3, right).

The model is validated by comparing its predictions against available experimental data as described in Methods section and shown in Figure 3.

### 3.2. Sensitivity Analysis

To evaluate the model response to parameter variations, a systematic sensitivity analysis of three model characteristics is performed. First, we assess the uncertainty of predicted progeny release kinetics caused by variations of the initial condition and model parameters in plausible ranges. Next, we identify the model parameters which have the most control over (1) the ratio of positive- to negative-sense viral RNAs and (2) the total number of virions released by 24 h post-infection.

#### 3.2.1. Uncertainty Analysis of the Progeny Release Kinetics

As part of model validation, we assess the uncertainty of progeny release kinetics caused by two factors. First, we predicted the changes in progeny trajectories obtained with initial condition $\left[V_{free}\right]\left(0\right)$ varied by ±50%, i.e., from 5 to 15 virions (Figure 4 (left)).

Next, we quantified the model uncertainty caused by the changes of model parameters (Figure 4 (right)). To this end, we randomly sampled a bunch of parameter combinations from the ranges of their permissible values as described in the Methods section. Model parameters presented in Table 2 can be either fixed ($f_{ORF1},f_{N},f_{SP},n_{N}$) or allowed to be varied. For most of the later parameters, their permissible ranges were estimated from the literature. For parameters $k_{transl},d_{ensodome},K_{NSP},d_{gRNA_{\left(-\right)}},d_{assembled}$, we estimated their ranges so that the overall uncertainty of progeny release was matched with about 10 to 10,000 virions (i.e., 1000-fold range) in accordance with Table 1. However, we additionally needed to narrow down the ranges of some parameters which were determined from literature, i.e., $d_{V},d_{NSP},k_{tr_{\left(-\right)}},d_{gRNA}$, so that the confidence region of progeny release at the end of replication cycle started from around ten virions but not from zero. As a result, this procedure allows us to consistently identify the plausible ranges for model parameters which ensure the robustness of the model behaviour in the range of its uncertainty that is matched to the level of uncertainty coming from current knowledge about the SARS-CoV-2 life cycle.

#### 3.2.2. Parameters Controlling the Ratio of Positive- to Negative-Sense vRNAs

Given the good agreement of the calibrated model with available data on the kinetics of positive- to negative-sense vRNA ratio (Figure 3), we asked which model parameters determine this ratio. To this end, we analyze parameter sensitivity towards the following distance $\Phi_{ratio}\left(p\right)$ between the ratio predicted by the model $r_{m}\left(t_{i}\right)$ and ratios $r_{exp_{\left(1\right)}}\left(t_{i}\right)$ and $r_{exp_{\left(2\right)}}\left(t_{i}\right)$ from the two-replicate experiment [24]:$$\begin{array}{cc} \Phi_{ratio}\left(p\right)= & \sum_{i}{\left(\log_{10}\left(r_{m}\left(t_{i}\right)\right)-\log_{10}\left(r_{exp_{\left(1\right)}}\left(t_{i}\right)\right)\right)}^{2}+\\ & \sum_{i}{\left(\log_{10}\left(r_{m}\left(t_{i}\right)\right)-\log_{10}\left(r_{exp_{\left(2\right)}}\left(t_{i}\right)\right)\right)}^{2},r_{m}\left(t\right)=\frac{\left[gRNA_{\left(+\right)}\right]\left(t\right)+\left[gRNA\right]\left(t\right)}{\left[gRNA_{\left(-\right)}\right]\left(t\right)}. \end{array}$$

The nonzero normalized sensitivity indices towards $\Phi_{ratio}$ are shown in Figure 5. The following parameters have the largest effect on the discrepancy between the modelled and experimentally obtained ratio of positive- to negative-sense vRNA:threshold number of [NSP] enhancing vRNA transcription,translation rate of non-structural proteins,rates of fusion and uncoating,replication rate of positive sense RNAs.

#### 3.2.3. Predicting Novel Antiviral Targets That Control Progeny Production

The calibrated model can be used to predict the sensitivity of SARS-CoV-2 production by an infected cell to variations of the rates of underlying biochemical processes. To identify the prospective antiviral targets, we analyze parameter sensitivity towards virus progeny production. To this end, we use two functionals: (a) total number of released virions during T=24 h post-infection, $\Phi_{progeny}\left(p\right)=\int_{0}^{T}k_{release}\left[V_{assembled}\right]dt\approx 280$ virions, and (b) the area under the curve type of metric for released virions during the same period, $\Phi_{progeny}^{AUC}\left(p\right)=\int_{0}^{T}\left[V_{released}\right]dt\approx 1208$ virions × hours. The sensitivity values, normalized by parameter values, are presented in Figure 6 and Figure 7. The left part of the figures ranks the sensitivity indices for the model parameters that negatively impact the net SARS-CoV-2 production when their values are increased. The right bar plot ranks the sensitivity indices for the parameters which positively affect the net virus production with their increasing values. It follows that the three parameters having the largest sensitivity indices towards both functionals are the following ones:degradation rate of positive sense vRNAs in cytoplasm (negative effect),threshold number of [NSP] enhancing vRNA transcription (negative effect),translation rate of non-structural proteins (positive effect).

## 4. Discussion

In this work, we have developed and calibrated a deterministic model of the SARS-CoV-2 life cycle at the level of the infected cell. The mathematical model describes the intracellular biochemistry underlying the replication of the virus. The model can be used as a part of multiscale models of the within-host SARS-CoV-2 infection, which are expected to provide a quantitative analytical tool to reveal complex pathogenetic mechanisms of COVID-19 following a systems approach [3].

The developed model predicts the dynamics of intracellular replication cycle of SARS-CoV-2 in target cells. The high initial condition of free infectious virions used in the paper corresponds to in vitro experimental studies of viral replication cycle with high MOI. This facilitates comparisons of model predictions of the dynamics of new viral progeny with those that would be estimated in future experimental setups. In such high MOI conditions, the entry of virions is very rapid [2]. In vivo, however, cells can be infected with a wide ranging number of free virions. Moreover, the expression of ACE2 on epithelial cells of various types and in different organs varies substantially [23,52], which would result in different kinetics of the virus entry. Therefore, the density of ACE2 on a cell membrane should be considered in the models of infection spreading within the human host organism. Furthermore, the lower the MOI, the more prevalent are stochastic effects at the early replication stages that result in the heterogeneity of the overall dynamics [53]. Thus, to predict the virus transmission in vivo, the multiscale hybrid models need to be developed that would incorporate the stochastic description of the intracellular SARS-CoV-2 life cycle.

Although many studies have been performed to characterize various aspects of the SARS-CoV-2 life cycle, detailed kinetics data similar to those available for HIV-1 [54] do not exist yet. Hence, the data “… to characterize the emergence of the viral replication intermediates and their impact on the cellular transcriptional response with high temporal resolution” [54] are urgently needed. In COVID-19, the virus infects multiple target cells expressing ACE2 including type I and II pneumocytes, alveolar macrophages, monocytes, endothelial cells, and airway epithelial cells, particularly of the mucous glands. To progress further with the development of relevant multiscale mathematical models, all these cell types need to be characterized with respect to the SARS-CoV-2 life cycle. For reviews on experimental models which are developed to study SARS-CoV-2 replication in cultures of various target cells, we refer to [55,56,57,58].

The deterministic quantitative description of the SARS-CoV-2 life cycle in the model establishes a mechanistic basis for the development of a stochastic description of the process kinetics using the Monte Carlo framework. This is required as many of the replication stages are characterized by low numbers of reactants and hence, the impact of random effects on the reaction kinetics is strong. The insight provided by stochastic models of influenza, hepatitis, and HIV infections [12,59] can be utilized in a similar way for SARS-CoV-2 modelling.

In our model, we do not consider spatial and transport aspects of the SARS-CoV-2 replication in the infected cell. An example of a stochastic approach to modelling the intracellular spatio-temporal dynamics of HIV-1 replication describing the microtubule transport of viral components is provided in [60]. In the case of SARS-CoV-2, adding the spatial dimension to the model would include the creation of replication compartments (e.g., double-membrane vesicles), intracellular trafficking (e.g., via the secretory pathway, and the molecular interactions with host proteins and innate immune responses [17]. For example, ORF3b, ORF6, and N proteins are known to interfere at multiple levels to inhibit the IFN-signalling pathway. The accumulation of viral factors leading to infected cell death via various mechanisms (apoptosis, necrosis, and pyroptosis) are not considered. These all require further analysis and strongly depend on the availability of appropriate quantitative data.

The developed mathematical model does not consider full details of subgenomic RNAs transcription and translation processes, and hence, the interaction of the encoded viral proteins with the intracellular defence mechanisms. The model needs to be linked to the cell physiology by considering the impact of virus infection on the codon usage, the cell metabolism, and the infected cell fate.

At present, there are no effective therapies against SARS-CoV-2 infection and computational modelling is used to assist with drug repurposing attempts [27]. Here, we have used our calibrated model to predict optimal targets for antiviral therapy. For this, all model parameters were ranked according to their contribution to the production of new virions by an infected cell using local sensitivity analysis. Parameters with the greatest negative and positive effects on virus progeny generation were identified. The three most influential parameters are (i) degradation rate of positive sense vRNAs in cytoplasm (negative effect), (ii) threshold number of non-structural proteins enhancing vRNA transcription (negative effect), and (iii) translation rate of non-structural proteins (positive effect). These parameters regulate the kinetics of various stages of the SARS-CoV-2 life cycle. Overall, systematic analyses of virus replication cycles with mathematical models of increasing level of detail should provide an efficient and rational way to define novel antiviral targets for therapies.

A quantitative mathematical description of the SARS-CoV-2 life cycle is a necessary step in gaining a predictive understanding of the regulatory mechanisms underlying the molecular biology of the virus and defining potential targets for inhibitors of the virus replication. Indeed, the sensitivity analysis of the model enables the prediction of potential targets for further experimental consideration, thus assisting the drug discovery or repurposing efforts. In fact, the derived model identifies such targets. The in silico model provides an analytical tool to examine the effects of certain virus mutations (e.g., the increase in affinity to ACE2 receptor) or combinations of multiple mutations on the viral progeny and drug resistance in advance of experiments. Finally, the model of SARS-CoV-2 replication being properly expanded in a question-driven manner could assist in gaining mechanistic insights into the robustness/fragility of intracellular defence mechanisms and the net outcomes of multi-modal therapeutic impacts which affect the specific stages of virus life cycle and the infected cell physiology.

## Figures and Tables

**Figure 1 viruses-13-01735-f001:**
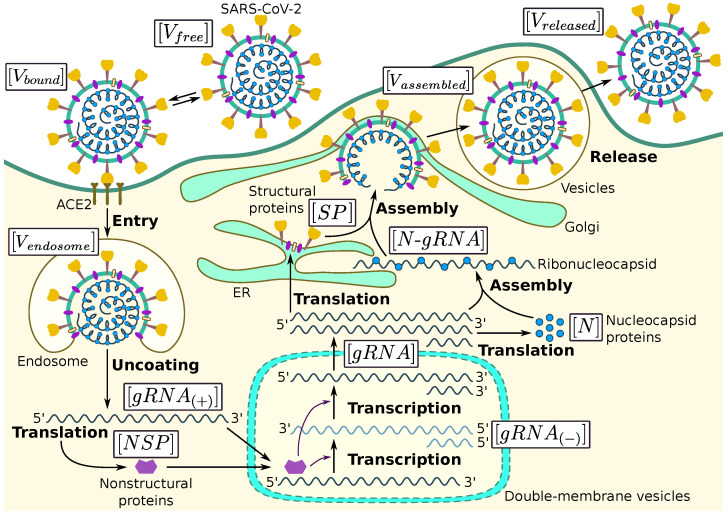
Biochemical scheme of the SARS-CoV-2 replication cycle.

**Figure 2 viruses-13-01735-f002:**
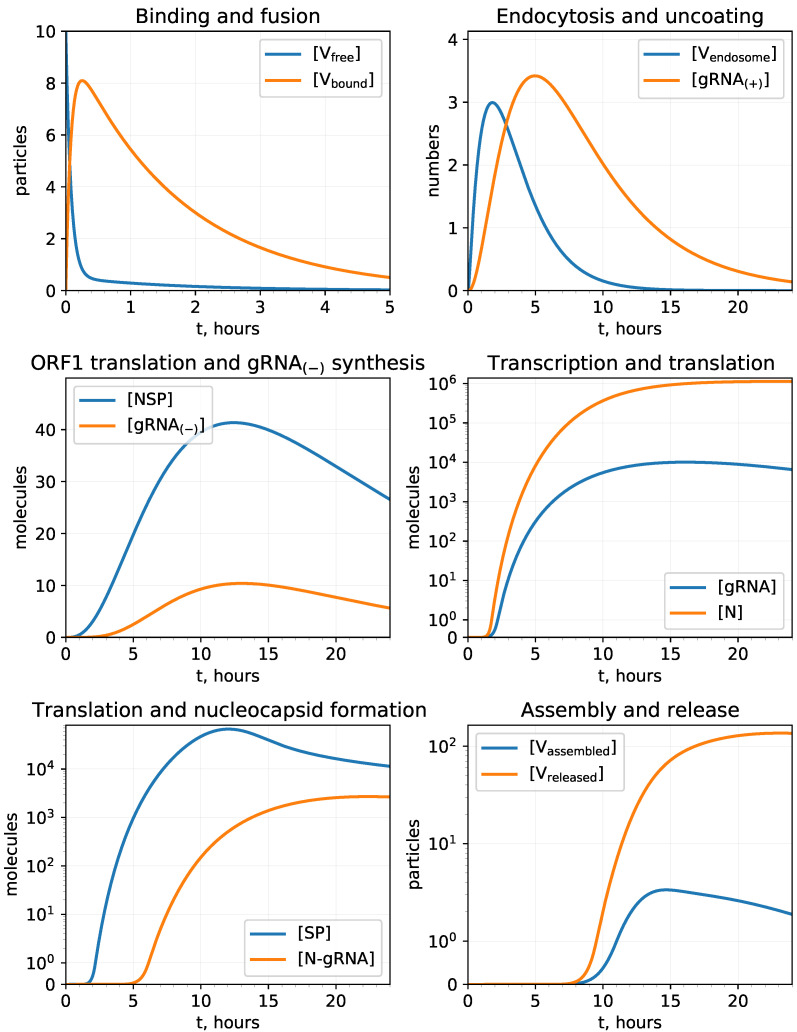
Reference model solution with parameters estimated in Table 2, $\left[V_{free}\right]\left(0\right)=10$. It predicts the kinetics of the viral replication intermediates. The upper row describes the cell entry state variables, i.e., (**upper left**) the free virions outside the cell membrane and the virions bound to ACE2, and (**upper right**) virions in endosomes together with single strand positive sense genomic RNA. The middle row specifies the kinetics of negative sense viral genome transcription and translation of non-structural proteins (**middle left**) followed by the positive sense genomic RNA transcription and the translation of N-protein (**middle right**). The bottom row displays the translation of the structural proteins and the ribonucleocapsid molecules formation (**bottom left**) resulting in creation of assembled virions in endosomes and the final release of virions (**bottom right**).

**Figure 3 viruses-13-01735-f003:**
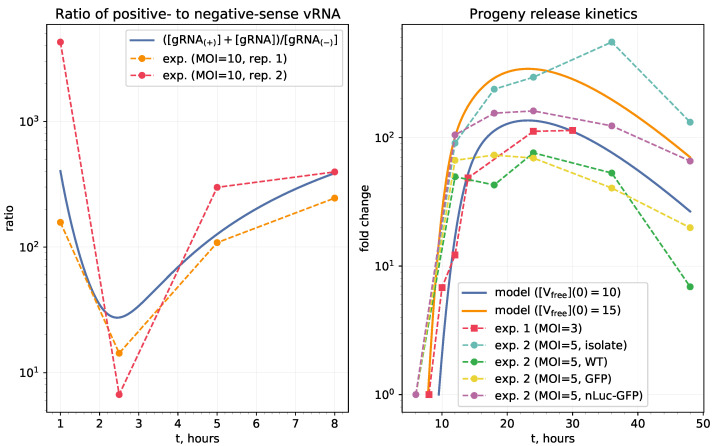
Validation of the calibrated model against available experimental data. (**Left**) ratio of positive- to negative-sense vRNA predicted by the model and measured in MHV-infected cells at MOI = 10 in [24]. (**Right**) SARS-CoV-2 progeny release kinetics in Vero E6 cells from [23,25] expressed in fold changes of the titer levels in cell culture from the start of their exponential growth. For model predictions, the moment of release onset was defined as the moment $t_{s}$ at which $\left[V_{released}\right]\left(t_{s}\right)=1$. The initial conditions were chosen to match the MOI used in experiments, i.e., so that the maximum number of uncoated $[gRNA_{\left(+\right)}]\approx$ MOI.

**Figure 4 viruses-13-01735-f004:**
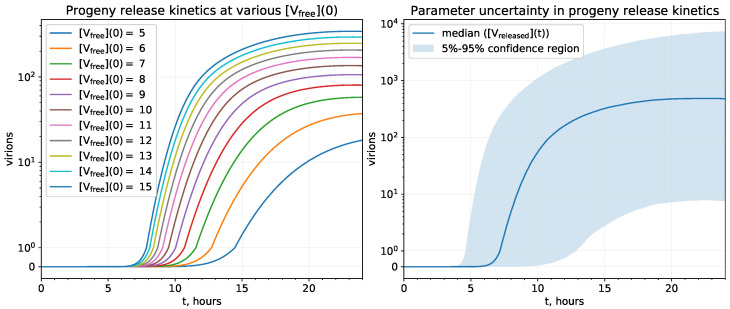
Uncertainty in the model output (progeny release kinetics). (**Left**) uncertainty associated with variation of the initial condition $\left[V_{free}\right]\left(0\right)$. (**Right**) uncertainty associated with variation of model parameters in the ranges specified in Table 2.

**Figure 5 viruses-13-01735-f005:**
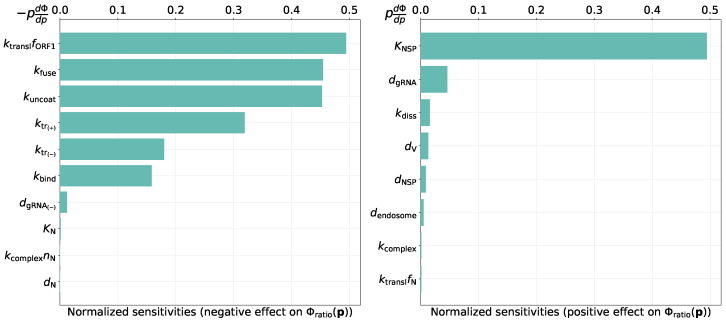
Local normalized sensitivity indices having negative (**left**) and positive (**right**) effects on the objective function $\Phi_{ratio}\left(p\right)$, i.e., on discrepancy between model and experiment.

**Figure 6 viruses-13-01735-f006:**
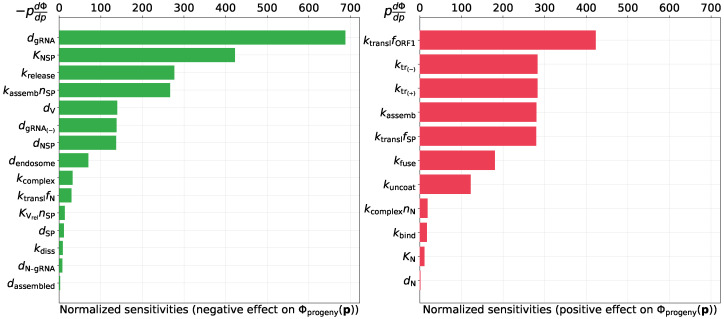
Local normalized sensitivity indices having negative (**left**) and positive (**right**) effect on the total number of released virions $\Phi_{progeny}\left(p\right)$.

**Figure 7 viruses-13-01735-f007:**
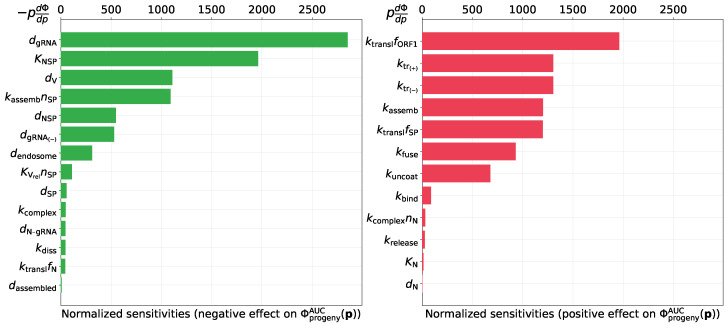
Local normalized sensitivity indices having negative (**left**) and positive (**right**) effect on the area under the curve for released virions $\Phi_{progeny}^{AUC}\left(p\right)$.

## Data Availability

Not applicable.

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
