# Peer review of "Intracellular Life Cycle Kinetics of SARS-CoV-2 Predicted Using Mathematical Modelling"

_viruses, 2021, doi:10.3390/v13091735_

Round 1
Reviewer 1 Report
The is an ambitious effort aimed at providing, for the first time since the COVID-19 pandemic arose, a detailed kinetic description of the major replication stages of SARS-CoV-2. To this end the authors performed an exhaustive survey of the available relevant literature pertaining to this and other corona viruses and formulated a plausible deterministic mathematical model summarizing the existing knowledge on the kinetics of productive infection of individual cells. A comprehensive sensitivity analysis allowed the authors to rank the model parameters according to their impact on the
net secretion of virions by the infected cell. The most influential parameters are considered as prospective targets for antiviral drugs. The considerable uncertainty with regard to the range of several molecular species (the model's variables) imposes corresponding uncertainty about the model's predictions, but the qualitative ranking may still be valid. Besides, the present results would be refined as more quantitative data become available, so that "systematic
analyses of virus replication cycles with mathematical models of increasing level of detail should provide an efficient and rational way to define novel antiviral targets for therapies."
General comments:
- I would suggest adding a methodological paragraph to the Introduction, explaining how the model was "calibrated". It might be explained that, given the experimental uncertainties, calibration did not amount to a rigorous parameter determination, but rather resulted from an iterative process of parameter guessing and constraining the choice progressively by requiring consistency with all available data. These data largely consisted of previously estimated numerical values or numerical ranges of the molecular species variables at key time points, such as the beginning and ending of the infection cycle or of intermediate steps (with a reference to Table 1).
- With the addition of such a paragraph, there would be no need to refer repeatedly to the previously established "quantitative characteristics" of the variables listed in Table 1 (unless additional information about those is considered useful), so that the discussion of parameter choices in the Results may be focused on consistency and self-consistency constraints. This in my view would simplify the discussion.
Specific comments:
Line 44: should be "...prospective..."
Line 48: should be "Figure 1" at the end of the line
Equation 3: should be Vbound (instead of Vbind)
Line 70: delete "of" from "fail to release of the viral genome"
Lines 75-79: too vague; reformulate these two sentences to explain more specifically the main kinetic features, e.g., in the form: "The model was calibrated to reproduce (1) ..., (2) ...., and (3) ....
Equation 5: is it indeed gRNA(+) in the first term on the r.h.s. (translation rate of NSP) rather than gRNA? See my next comment.
Lines 92-93. This sentence is a bit confusing, prompting my previous comment. "It is taken into account in equations (5) and (6) that the newly synthesized genomes participate in the generation of more NSPs." This appears to suggest a positive feedback loop, but the above equations do not represent such a loop. (Did the authors consider such a mechanism?) If gRNA(+) is correct, the sentence should be erased.
Lines 94-95: "with a general range of about 1 to 104
copies/mRNA h−1 [21]" can be deleted once the additional text I have proposed in my general comment (1) above (or similar text) is included (see also general comment number (2).
Lines 98-100: I would replace simply with something like: This result of our calibration procedure is consistent with the finding that the life time of most proteins is just few hours [21].
Lines 102-103: The sentence: "This is consistent with the transcription rate of mRNAs which ranges from 1 to 100 copies per hour according to [21]" can now be erased (see above).
Line 143: Similarly, the sentence "with a general range of about 1 to 104 copies/mRNA h−1[21]" can be erased
Line 187: "depends on"
Table 2, second line: erase "free"
Line 185: include MOI in the list of abbreviations
Lines 189-190: "...sensitivity... to" instead of "...dependence...on..."
Lines 213-216: Too vague. Please reformulate and simplify.
(e.g., the MOI, [Vfree], or the the negative and positive RNAs); previous estimation
Reviewer 2 Report
In this work by Grebennikov et al, they develop a model of the kinetics of the virus intracellular life cycle and then analyze the model to assess what are the most significant interactions that regulate virus reproduction. The model construction is well justified, however, there are concerns about the model analysis and the parameter uncertainties that limit the impact of the study. Please see below:
Comments:
- There is no “Methods” section and it needs to be added. Typically the model description would be included there and only the descriptions of the simulation results are discussed in the Results section. Also, the authors do not describe the software that was used or what packages and dependencies there are. These are requires materials.
- The model code must be deposited to support peer review. Code is typically deposited on Biomodels.net or the team’s GitHub. Without the code, it is not possible to fully review the model.
- Table 2: The authors do a nice job digging into the literature to attain initial parameter estimates. However, the estimates come from different viruses (sometimes flu, sometimes SARS-CoV-2) and different cell types. And it is well known in bio systems that these parameter values have large error ranges. To have any confidence in the model, the authors should perform an analysis of the model’s robustness by simulating the model responses across probably a few millions combinations of parameters. And those parameters should be selected by sampling the error ranges identified in the experimental study, or, few when using non-SARS-CoV-2 data, over a range of magnitude. This will tell us for what range of parameters the model is reasonable and help identify experimental conditions for model verification.
- It is unclear is this model is meant to be an in vitro model or an in vivo model. In figure 2, virus fusion happens very rapidly and I’d guess this is only relevant to high MOI in vitro experiments.
- Figure 2 caption needs more information. Why stop the simulation at 24 hours?
- Also, what data is available to justify the dynamics shown? Viruses is geared to experimentalists. Any data from the literature that justify these plots should be added to the figure, with error bars, and the source cited.
- Please add more analysis of the model. E.g. What conditions restrict burst size (Vassebled), what controls the ratio between gRNA_ and gRNA+, and so on. These types of simulations and analyses can add validity to the model.
- Figure 4 is not necessary. It might be better to just include the descriptions of the parameters in Figure 3 in addition to the parameter variable name.
- Line 209: “first mathematical model describing the intracellular biochemistry underlying replication of the virus”. The paper cites an already published SARS-CoV-2 model….Wang, S.; Pan, Y.; Wang, Q.; Miao, H.; Brown, A.N.; Rong, L. Modeling the viral dynamics of SARS-CoV-2 infection. Please explain why this model is unique.
- Line 213 “Menten-type description to account for 3 aspects of virion assembly:” This content belongs in the model development section.
- The discussion in incomplete. It should be better oriented towards the experimentalists. What experiments need to be done to justify this model? What cell types in the lung is it best gear toward? What are the model’s current major shortcoming and how can they be addressed?
Reviewer 3 Report
In the manuscript entitled “Intracellular life cycle kinetics of SARS-CoV-2 predicted using mathematical modelling”, Grebennikov et al. present a mechanistic model to account for SARS-CoV-2 replication.
The paper is deals with a very important topic, especially given the current situation, creates a beautiful model, and I enjoyed reviewing it. However, it was complicated to read the methods section and I would suggest some modifications to make the paper more understandable. Upon addressing them, I recommend publication.
- Introduction: please extend the discussion of mechanistic mathematical models for viruses.
- Section 2.2 is a crucial part of the paper and it needs to be explained better. Perhaps the authors could the authors could add a description of the experimental data used to make the model, and an explanation on how it was generated. Otherwise the results of Figure 3 are hard to follow.
-Small comment on style of display Equation 7 and 8. It would be easier to read the equations if the term [gRNA] in Equation 8 was moved to equation 7, and f_complex be defined as [N]/([N]+K_N).
-Modify accordingly equations 9,10, 11, 12, 13.
-Figure 3 appears in the text before Figure 2.
-Line 308: is 6000 an arbitrary number?
-In the discussion, the authors mention that their model does not consider spatial and transport aspects of the virus during replication, and they discuss the necessary biological elements to be modelled. I miss a reference where this has been mathematically modelled in the context of another virus.
-A weakness of the manuscript is that the authors do not derive a strong biological result from it, which is totally fine since the goal of the paper is to produce a model. However, a paragraph in the Discussion summarizing future lines of research to achieve this, could make it much stronger, and it could motivate biologists to use this type of modelling for their work.
